# Cluster Analysis of Subjective Shoulder Stiffness and Muscle Hardness: Associations with Central Sensitization-Related Symptoms

**DOI:** 10.3390/medicina59101831

**Published:** 2023-10-14

**Authors:** Natsuna Komoto, Hanako Sakebayashi, Naoto Imagawa, Yuji Mizuno, Ibuki Nakata, Hayato Shigetoh, Takayuki Kodama, Junya Miyazaki

**Affiliations:** Department of Physical Therapy, Faculty of Health Science, Kyoto Tachibana University, 34 Yamada-cho, Oyake, Yamashina-ku, Kyoto 607-8175, Japana903020031@st.tachibana-u.ac.jp (N.I.); kodama-t@tachibana-u.ac.jp (T.K.); j-miyazaki@tachibana-u.ac.jp (J.M.)

**Keywords:** shoulder stiffness, subjective symptoms, muscle hardness, subgroup, psychological factors

## Abstract

*Background and Objectives*: Understanding the relationships between subjective shoulder stiffness, muscle hardness, and various factors is crucial. Our cross-sectional study identified subgroups of shoulder stiffness based on symptoms and muscle hardness and investigated associated factors. *Materials and Methods*: measures included subjective stiffness, pain, muscle hardness, and factors like physical and psychological conditions, pressure pain threshold, postural alignment, heart rate variability, and electroencephalography in 40 healthy young individuals. *Results*: Three clusters were identified: Cluster 1 with high stiffness, pain, and muscle hardness; Cluster 2 with low stiffness and pain but high muscle hardness; and Cluster 3 with low levels of all factors. Cluster 1 had significantly higher central sensitization-related symptoms (CSS) scores than Cluster 2. Subjective stiffness is positively correlated with psychological factors. *Conclusions*: our results suggest that CSS impacts subjective symptom severity among individuals with similar shoulder muscle hardness.

## 1. Introduction

In recent years, the prevalence of shoulder stiffness has surged due to extended smartphone use and deskwork, with it being among the top reported symptoms in both men and women [1]. Shoulder stiffness is characterized by discomfort, dull pain, heaviness, and muscle tension in the neck, scapulae, and shoulder regions [2]. While some shoulder stiffness can be traced to underlying diseases, primary shoulder stiffness occurs without a specific cause and has links to factors like poor posture, psychological stress, and aging [3,4,5].

Muscle hardness was used as an objective evaluation method for shoulder stiffness. Higher trapezius muscle hardness and tenderness at the symptomatic site have been re-ported in individuals with shoulder stiffness [6,7,8]. Some research suggests that increased trapezius muscle hardness correlates with subjective shoulder stiffness symptoms [9], while others disagree [10]. Moreover, discrepancies exist regarding the roles of posture, impaired blood flow, and other factors in muscle hardness [11,12,13,14].

The emotional component of pain in individuals with shoulder stiffness has an association with conscious subjective stress [15]. In addition, shoulder stiffness and psychological stress are included as central sensitization-related symptoms (CSS), suggesting a link to the central nervous system. Some studies suggest relationships between unconscious psychological stress and changes in the autonomic nervous system (ANS) and brain activity [16,17,18,19,20]. However, no study has simultaneously examined these psychosomatic indices.

The current literature reveals an inconsistency in the relationship between objective findings (muscle hardness) and subjective symptoms of shoulder stiffness [8,9,10]. To address this, we propose the following four hypothesized subgroups based on differences in the severity of subjective symptoms and muscle hardness:High subjective stiffness and pain with high muscle hardness;Low subjective stiffness and pain with low muscle hardness;High subjective stiffness and pain with low muscle hardness;Low subjective stiffness and pain with high muscle hardness.

Additionally, we hypothesized that the differences between subgroups (subgroups 1 and 2) with similar muscle hardness but varying levels of subjective discomfort would be related to daily conscious psychological stress, CSS, or nonconscious psychological stress, which EEG or ANS indicates. On the other hand, we hypothesized that the differences in muscle hardness were associated with physical factors, such as posture alignment. This study aimed to verify the existence of subgroups of shoulder stiffness based on subjective symptoms and muscle hardness and to clarify the comprehensive factors associated with increased subjective symptoms and muscle hardness.

## 2. Materials and Methods

### 2.1. Participants

This study involved 40 healthy young individuals (average age: 20.7 ± 0.6 years; 20 males and 20 females). Participants with histories of numbness or musculoskeletal disorders of the neck, upper limbs, or back were excluded. This study excluded secondary shoulder stiffness, which was induced by a specific disease. In addition, participants were asked to avoid caffeine and alcohol and maintain their regular sleep duration before the measurement. The study was approved by the Kyoto Tachibana University Ethics Committee (approval number: 23-03), and the participants provided oral consent after being informed about the study. This study was conducted between 1 April and 30 April 2023.

### 2.2. Procedure

Initially, we administered a questionnaire to assess physical and psychological factors related to shoulder stiffness. We then measured the participants’ height and weight and recorded EEG and ANS activity in a relaxed static sitting position. Subsequently, we evaluated postural alignment, muscle hardness, and pressure pain threshold (PPT). The study was conducted in a quiet environment at a room temperature of 23 °C, with participants wearing short-length clothing. For seated measurements, the participants used a chair that allowed their feet to touch the ground and were instructed not to lean on the backrest. All evaluations were performed between 1 pm and 6 pm within one day.

### 2.3. Subjective Symptoms of Shoulder Stiffness

We assessed the severity of the participants’ subjective stiffness and pain related to shoulder stiffness using an 11-point Numerical Rating Scale (NRS) ranging from 0 (none) to 10 (extreme).

### 2.4. Lifestyle Related to Shoulder Stiffness

Based on a previous study [4,5], we asked the participants about their average daily hours of VDT work, average sleep duration, and weekly exercise hours for light, moderate, and high-intensity loads. Exercise load definitions were based on the International Physical Activity Questionnaire (IPAQ) [21].

### 2.5. Central Sensitivity Syndromes (CSS)

We used the Central Sensitization Inventory (CSI) to evaluate the CSS. CSS is a health-related symptom, such as headache, muscle stiffness, sleep disorder, and fatigue, based on central sensitization. The CSI, which consists of 25 items, has been validated for reliability and validity and helps diagnose CSS [22].

### 2.6. Psychological Factors

We assessed participants daily conscious psychological stress levels using the NRS and participants’ mood at the time of measurement using the Two-Dimensional Mood Scale (TDMS) [23]. The TDMS measures psychological state based on eight items assessing arousal, comfort, stability, and activity levels. This has been reported to be highly reliable and valid [23].

### 2.7. EEG

Psychological stress, especially cognitive and emotional states, plays a role in the subjective experience of shoulder stiffness. EEG provides an objective measure of these psychological states. We conducted EEG measurements to use them as objective indicators of psychological stress, including cognitive and emotional states. An EEG (Biosignalplux, Plux, Inc., Lisbon, Portugal) with a sampling frequency of 1000 Hz was used. Electrodes were placed at Fp1 and Fp2, associated with the prefrontal cortex, which is responsible for higher cognitive functions such as thinking, behavior, motivation, attention, and emotional processing [24,25,26]. The participants were instructed to sit quietly with their eyes open and stare blankly at a considerable mark on the wall.

MATLAB R2023a was used to analyze brainwaves. The total measurement time was 5 min and was filtered with a 1–30 Hz bandpass filter. Frequency analysis was performed using the Fast Fourier Transform (FFT) in 1000-point segments every 5 s. The absolute power values of alpha (α) waves (8–12 Hz) and beta (β) waves (13–25 Hz) were calculated. The power values in each frequency band were log-transformed because they exhibited an exponential variation [27]. This study calculated log α, log β, and log α/β at Fp1 and Fp2 as physiological indicators.

### 2.8. ANS Activity (Heart Rate Variability)

The ANS, responsible for bodily functions like heart rate, can provide insights into the overall physiological stress levels, potentially offering an understanding of the relationship between shoulder stiffness and systemic responses. ANS activity was measured simultaneously with the EEG using a plethysmography sensor (BioSignalplux, Plux, Inc., Lisbon, Portugal). The sensor was attached to the index finger of the right hand of each participant. Plethysmography captures volume changes in blood vessels caused by the heart pumping blood. The sampling frequency was set to 1000 Hz. OpenSignals software (v. 2. 2. 3a) was used to analyze the volumetric pulse waves. After measuring the waves, frequency analysis was used to calculate sympathetic and parasympathetic nervous system activity using FFT. Absolute power values in the high-frequency band (HF: 0.15–0.40 Hz) and low-frequency band (LF: 0.05–0.15 Hz) were log-transformed as they exhibited exponential variation, and the LF/HF ratio was also calculated for statistical analysis [16]. An increase in HF power values indicates an increase in parasympathetic nervous activity [16], whereas an increase in the LF/HF ratio indicates an increase in sympathetic nervous activity [16].

### 2.9. Posture Alignment Assessment

To evaluate the impact of postural alignment on shoulder stiffness, we assessed the participants’ posture during quiet sitting. They were instructed to look straight ahead and sit without leaning on a backrest. Markers were placed on the earlobe and seventh cervical vertebra (C7), and photographs were taken from 2.5 m to the participant’s left. Posture alignment was analyzed using the VisionPose application, which employs deep learning for posture estimation. VisionPose is utilized to measure postural alignment [28]. The spinal angle and head position were calculated as postural alignment indicators during quiet sitting. The spinal angle was determined using the midpoint of the shoulders, spine, and waist center. The head position was assessed using the Craniovertebral Angle (CVA), which measures the angle between the horizontal line passing through C7 and the line connecting C7 to the earlobe [29].

### 2.10. Muscle Hardness

Muscle hardness directly ties to the physical manifestation of shoulder stiffness, offering an objective measure of the condition itself. We used a muscle hardness meter (NEUTONE TDM-Z1, TRY-ALL, Chiba, Japan) to measure the hardness of the upper trapezius muscle fibers on both sides. The measurement location was based on a previous study that identified the midpoint of the line connecting the seventh cervical vertebra and the left and right acromions as the upper trapezius muscle at the measurement point [10]. The participants were seated and instructed to extend their backs, look straight ahead, and avoid moving as much as possible. Measurements were performed five times, and the average values of the left and right sides were calculated as muscle hardness values. The intraclass correlation coefficient (ICC) for muscle hardness measurement was 0.99, indicating good reliability.

### 2.11. PPT

Shoulder stiffness may result from localized sensitivities, and the PPT provides a means to assess the sensitivity or pain response in the shoulder region. After measuring muscle stiffness, we assessed the participants’ PPT using a digital force gauge (model RZ-10, Aikoh Engineering, Osaka, Japan). The measurement sites and conditions were the same as those used to measure muscle hardness [30]. A gauge was vertically applied to the skin surface of the shoulder and pressed at a force of 5 N/s. Measurements were performed five times, and the average values of the left and right sides were calculated as the PPT. The ICC for the PPT measurements in this study was 0.99, indicating good reliability.

### 2.12. Statistical Analysis

To classify the subgroups based on subjective and objective shoulder stiffness indicators, we conducted cluster analysis (Ward’s method) using standardized (Z-scored) values for subjective shoulder stiffness severity, pain severity, and muscle stiffness. A dendrogram was used to classify the patients into three subgroups. We performed logarithmic transformations for non-normally distributed continuous variables to compare variables between clusters, followed by one-way analysis of variance (ANOVA) and multiple comparisons (Bonferroni correction). Chi-square analysis was conducted for categorical variables such as sex. To examine whether factors related to subjective shoulder stiffness, pain, and muscle stiffness differed according to the presence or absence of shoulder stiffness, we used Pearson’s correlation analysis for the entire participant group and subgroups with and without shoulder stiffness. The significance level was set at 5%. We used R version 4.2.3 for statistical analysis.

## 3. Results

### 3.1. Results of Cluster Analysis and Multiple Comparisons Based on Subjective Shoulder Stiffness, Subjective Pain, and Shoulder Muscle Stiffness Values

Table 1 presents the results of the measured items for all participants and each cluster. Figure 1 shows a scatterplot of standardized (Z-scored) subjective shoulder stiffness, subjective pain, and shoulder muscle stiffness values. Cluster analysis classified participants into Cluster 1, with high subjective pain, stiffness, and muscle stiffness; Cluster 2, with low subjective pain and stiffness but high muscle stiffness; and Cluster 3, with low values across all variables. Multiple comparisons revealed significant differences in shoulder muscle hardness between Custers 1 and 3 (*p* < 0.01) and Clusters 2 and 3 (*p* < 0.01). Significant differences were also found in subjective stiffness and pain between Clusters 1 and 2 (*p* < 0.01) and Clusters 1 and 3 (*p* < 0.01) (Figure 2).

These data showed three distinct clusters with varied degrees of subjective and muscle stiffness and pain. Interestingly, Cluster 2 had low subjective readings but still showed high muscle stiffness, indicating that subjective feelings do not always align with objective stiffness measures. The significant differences between Clusters 1 and 3 and Clusters 2 and 3 suggest there might be underlying factors differentiating these groups.

The y-axis indicates the Z-score of the muscle hardness. The x-axis indicates the Z-score of subjective stiffness. The bubble size indicates the amount of pain. The orange circle indicates Cluster 1, the blue circle indicates Cluster 2, and the gray circle indicates Cluster 3.

### 3.2. Comparison of Questionnaire Evaluation Items across Clusters

There were no significant differences in basic demographics (age, sex, height, weight, and BMI) or physical stress indicators (VDT work time, sleep duration, and exercise habits) between the clusters according to multiple comparisons and chi-square tests. The CSI scores revealed a significant difference between Clusters 1 and 2 (*p* < 0.01) but no significant differences between Clusters 1 and 3 or between Clusters 2 and 3. Furthermore, no significant differences were found across clusters for psychological stress indicators, including daily conscious psychological stress, TDMS comfort, arousal, stability, and activity levels.

While there were no differences in basic demographics or physical stress indicators across clusters, the notable difference in CSI scores between Clusters 1 and 2 implies that cognitive factors might influence subjective interpretations of stiffness and pain. The lack of differences in psychological stress indicators suggests that these might not directly influence the participants’ stiffness and pain perceptions.

### 3.3. Comparison of Posture, Pressure Pain Threshold, ANS Activity, and Brainwave Activity across Clusters

Multiple comparison analyses revealed no significant differences between clusters in posture-related factors (CVA and spinal angle) or PPT. Furthermore, no significant differences were observed across all clusters for the ANS and brainwave activity, which were assessed as objective measures of unconscious psychological stress.

The findings indicate that posture, PPT, ANS activity, and brainwave activity do not substantially contribute to differentiating the clusters. These results suggest that while these factors might have been suspected contributors, they do not significantly influence the clusters’ perceptions and objective measures of stiffness and pain.

### 3.4. Correlation Analysis of Subjective Shoulder Stiffness, Pain, and Muscle Stiffness with Other Variables

Table 2 presents the results of the correlation analyses of subjective shoulder stiffness, pain, and muscle stiffness with other factors. For all participants, significant positive correlations were found between subjective stiffness and pain (r = 0.71, *p* < 0.01), CSI scores (r = 0.59, *p* < 0.01), and daily conscious psychological stress (r = 0.44, *p* < 0.01). Subjective pain showed significant positive correlations with subjective stiffness (r = 0.71, *p* < 0.01) and CSI score (r = 0.48, *p* < 0.01) and a significant negative correlation with the Fp1 α/β ratio (r = −0.33, *p* = 0.04). No significant correlation was observed with shoulder muscle hardness.

When analyzed according to the presence or absence of shoulder stiffness, participants with shoulder stiffness exhibited significant positive correlations between subjective stiffness and subjective pain (r = 0.41, *p* = 0.048), shoulder muscle stiffness (r = 0.45, *p* = 0.03), daily conscious psychological stress (r = 0.50, *p* = 0.01), and CSI scores (r = 0.42, *p* = 0.04). A significant negative correlation was observed with TDMS stability (r = −0.42, *p* = 0.04). Subjective pain only showed a significant positive correlation with moderate-load exercise (r = 0.65, *p* < 0.01). Shoulder muscle stiffness had a significantly positive correlation only with subjective stiffness (r = 0.45, *p* = 0.03). Conversely, participants without shoulder stiffness exhibited a significant negative correlation between muscle stiffness and parasympathetic nervous activity (HF power) (r = −0.61, *p* = 0.01).

The strong positive correlation between subjective stiffness and pain indicates that individuals who feel stiffer also tend to feel more pain. This could suggest a common underlying factor or that the sensation of stiffness exacerbates pain perception. The positive correlation of these factors with daily conscious psychological stress and CSI scores may imply that cognitive and psychological states influence or are influenced by stiffness and pain perceptions. Additionally, the fact that subjective pain had a negative correlation with the Fp1 α/β ratio might indicate that certain brain activities are inversely related to pain perception. The difference in correlations among participants with and without shoulder stiffness might also suggest different underlying processes or influences for these two groups. Overall, the results emphasize the complexity of shoulder stiffness and pain perceptions and their interaction with cognitive and psychological states.

## 4. Discussion

In this study, we hypothesized that the participants could be classified into four subgroups based on subjective stiffness, pain, and muscle hardness. However, the analysis resulted in three subgroups: Cluster 1 had high levels of subjective stiffness, pain, and muscle hardness; Cluster 2 had low levels of subjective stiffness and pain but high muscle hardness; and Cluster 3 had low levels of all three factors. Multiple comparisons between subgroups revealed significant differences only in CSI scores, with Cluster 1 showing higher values than Cluster 2. No significant differences were observed in other conscious, physical, and psychological stress-related factors. Correlation analysis investigating factors related to subjective stiffness, pain, and muscle hardness showed associations between subjective stiffness, pain, the TDMS stability score, daily conscious psychological stress, and the CSI score in participants with shoulder stiffness. Subjective pain was suggested to be related to the Fp1 α/β ratio. Moreover, muscle hardness tended to be lower in participants without subjective shoulder stiffness when parasympathetic nervous activity was high.

In the present study, significant differences were observed in subjective shoulder stiffness and pain. Among the comprehensive factors related to conscious physical and psychological stress, only CSI scores showed significant differences, with no significant differences found in other factors. The CSI is a CSS questionnaire that assesses subjective symptoms, consisting of subfactors such as physical symptoms, emotional distress, headache, temporomandibular joint symptoms, and urological symptoms [31]. The CSI also contains items related to sleep disturbance, fatigue, and emotional distress, which are reportedly associated with muscle stiffness [32]. Furthermore, CSS has been reported to be associated with pain owing to the influence of psychological stress as a mediating factor [33]. One study reported a relationship between posture and shoulder stiffness [29]. However, this study suggests that conscious physical stress factors such as posture and physical activity are unrelated to subjective shoulder stiffness and pain. Furthermore, no significant differences were observed in the ANS or brainwave activity between Clusters 1 and 2. According to previous studies, subjective stiffness and pain are related to muscle hardness and ANS activity indicators such as the LF/HF ratio [34,35]. Brainwave activity, such as ANS activity, is sometimes used as an objective indicator of psychological stress. Brainwaves and ANS activity reflect unconscious stress [36], while questionnaire evaluations reflect conscious stress. This study did not observe any differences between brain waves and ANS activity. Therefore, it is suggested that CSS’s conscious physical and emotional stress factors are related to the differences in subjective shoulder stiffness and pain observed between Clusters 1 and 2. The findings that CSS is associated with subjective symptoms of shoulder stiffness suggest the importance of CSS-focused assessment and intervention.

Between Clusters 2 and 3, there were no significant differences in subjective shoulder stiffness or pain, both of which were low. However, there was a significant difference in muscle hardness. Previous studies reported that the hardness of the trapezius muscle does not directly reflect subjective shoulder stiffness, and the findings of this study are consistent with those of previous studies. Clusters 2 and 3 had low subjective shoulder stiffness and pain scores, suggesting that shoulder stiffness caused minimal conscious psychological stress. No significant differences were observed in factors related to psychological stress. The differences between clusters 2 and 3 were presumed to be influenced by conscious and unconscious physical stress factors. Previous studies have reported that forward head posture [29] and VDT work [37] can strain the neck and shoulder muscles. However, this study found no significant differences in the factors related to posture and physical activity, suggesting that differences in muscle hardness are not related to posture or physical activity. A previous study reported that the muscle hardness of the trapezius muscle is associated with blood flow velocity [38]. In addition to physical factors such as posture measured in this study, blood flow may have been related to muscle hardness. 

For Clusters 1 and 3, Cluster 1 showed significantly higher values for subjective shoulder stiffness, pain, and muscle hardness. Although there was no significant difference in CSI, Cluster 1 had more severe cases and higher CSI values than Cluster 3. No significant differences were found in other factors, suggesting that even if subjective shoulder stiffness, pain, and muscle hardness differed, there might be no differences in conscious/unconscious physical and psychological stress factors. Furthermore, previous studies have reported associations between blood circulation and shoulder muscle fatigue [13]. Factors not examined in this study, such as these, could also influence the differences in subjective shoulder stiffness, pain, and muscle hardness.

In this study, we found that CSS severity was associated with subjective symptoms. This study observed significant positive correlations among CSI scores, subjective stiffness, and pain. It has been reported that CSI scores are related to pain due to the influence of conscious psychological stress [33]. Additionally, a negative correlation was observed between the left Fp1 α/β ratio and pain. The left hemisphere is primarily involved in pleasant emotion processing [39], while the right hemisphere is primarily involved in arousal and unpleasant emotion processing. These findings suggest that individuals with higher subjective pain may have been affected by unconscious psychological stress mediated by CSS, with reduced pleasant emotions. A significant negative correlation was also observed between the stability score (TDMS) and subjective stiffness in individuals with shoulder stiffness. A lower stability score indicates an irritant and tense state, suggesting that individuals with higher tension may experience more substantial subjective stiffness due to irritation. A significant positive correlation was also observed between daily conscious psychological stress and subjective stiffness, indicating that subjective stiffness may be due to subjective conscious psychological stress, which is consistent with previous research [2]. Furthermore, this study found a negative correlation between the LF/HF ratio, sympathetic nervous system activity indicators, and pain in patients with shoulder complaints. An increased LF/HF ratio indicates increased sympathetic nervous system activity [16]. However, while previous research found a significant positive correlation between subjective stiffness, pain, and sympathetic activity [40], our study showed no correlation. These results suggest a balance between sympathetic and parasympathetic nervous system activities in individuals with shoulder stiffness. Some studies have argued that the LF/HF ratio does not necessarily indicate increased sympathetic nervous system activity [41]. Further investigation is required to understand its relationship with ANS activity. The current study’s comprehensive review of factors associated with shoulder stiffness suggests that CSS is related to subjective symptoms. CSS has a pathological basis in central sensitization. Our research results suggest that the pathogenesis of shoulder stiffness is also related to the pathophysiology of central sensitization, and it may be helpful for clinical evaluation and intervention.

To our knowledge, this is the first study to analyze unconscious psychological stress using brainwaves and ANS activity. Our results revealed that individuals with similar muscle hardness experienced different subjective symptoms, with CSS emerging as a related factor. Moreover, individuals who did not report subjective shoulder stiffness exhibited lower muscle hardness when the parasympathetic nervous system activity was high. Regarding brain wave activity, the entire study population demonstrated a tendency for subjective pain to decrease as relaxation increased owing to alpha wave activity in the left hemisphere, which indicates pleasure. Traditionally, interventions for shoulder stiffness have focused on physical factors such as posture [42]. However, our findings suggest that subjective symptoms are related to CSS and emotional factors, indicating that comprehensive management strategies that focus on cognitive-emotional aspects, such as cognitive-behavioral therapy, may be necessary.

This study had some limitations. First, we assessed the muscle condition using a muscle hardness meter. Previous studies have suggested that impaired blood flow to the shoulder muscles due to circulatory disorders could be a factor in shoulder stiffness [13]. Using muscle circulation as an assessment indicator, we identified new factors that affect subjective shoulder stiffness, pain, and muscle hardness. Secondly, our study included only healthy young adults. Previous research has indicated that shoulder stiffness symptoms and causes may vary depending on age and occupation [2]. Thus, it is necessary to expand the participant pool to include a broader range of ages and occupations. Third, it is uncertain whether the participants perceived “shoulder stiffness” negatively as a conscious psychological stressor. Using a questionnaire, we assessed mood states during measurement and daily conscious psychological stress, in addition to shoulder stiffness and pain. However, we could not evaluate whether participants with shoulder stiffness complained of feeling distressed due to the stiffness. Investigating the cognitive and emotional aspects of shoulder stiffness perceived as distress may help clarify the relationship between shoulder stiffness and conscious psychological stress. Fourth, it was unclear whether primary or secondary pain was included among the participants with shoulder stiffness in this study since the pain quality was not evaluated. In addition, site-specific questionnaire evaluation of shoulder stiffness was not possible. Fifth, the sample characteristics in this study might have affected the high and low levels of three factors of muscle hardness, subjective stiffness, and pain, as the severity related to these factors was classified based on Z-scores rather than absolute numerical criteria. Sixth, we evaluated the EEG at only Fp1 and Fp2. Fp1 and Fp2 have many different cognitive functions. Therefore, factors other than psychological stress may have influenced the EEG results in this study. Interestingly, the present study examined the pathophysiology of stiff shoulders from a comprehensive set of factors and suggested that CSS is related to subjective symptoms. However, other factors that may influence the pathophysiology of stiff shoulders are also considered, and whether the subjects suffered from stiff shoulders themselves may have influenced the study results. In the future, validating the results with individuals who perceive stiff shoulders as painful and including indices that could not be measured in this study is essential.

## 5. Conclusions

Our study established a connection between CSS and variations in the severity of subjective symptoms, even among participants with comparable shoulder muscle hardness. Furthermore, elevated conscious psychological stress and diminished brain wave activity indicating positive emotions, are notably linked with subjective symptoms in those suffering from shoulder stiffness. Our findings may help in the management of shoulder stiffness.

## Figures and Tables

**Figure 1 medicina-59-01831-f001:**
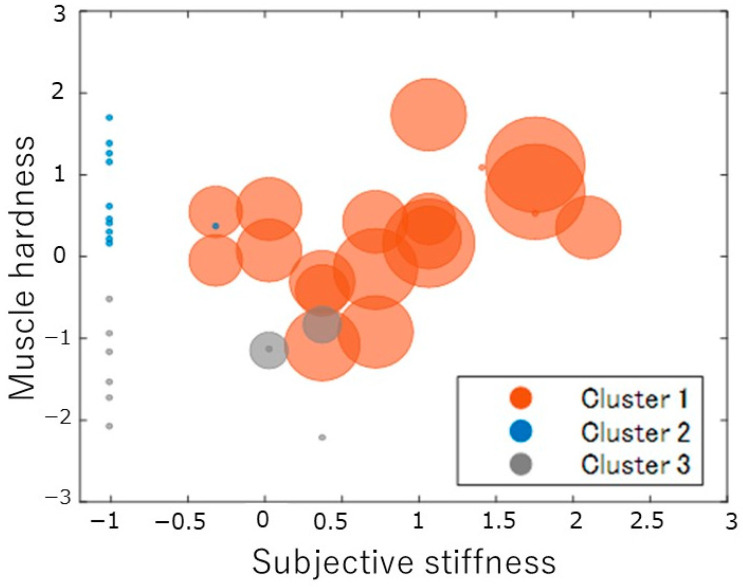
The bubble chart is based on muscle hardness and subjective symptoms.

**Figure 2 medicina-59-01831-f002:**
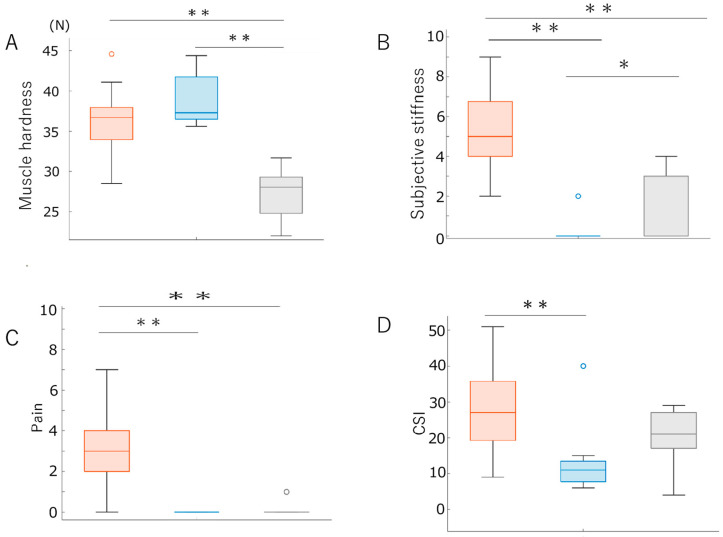
The difference in muscle hardness, subjective stiffness, pain, and CSI between clusters. (**A**) Differences in muscle hardness between clusters. (**B**) Differences in subjective stiffness between clusters. (**C**) Differences in pain between the clusters. (**D**) Differences between CSI. The orange, blue, and gray boxes indicate Clusters 1, 2, and 3, respectively. ** *p* < 0.01. * *p* < 0.05. CSI. Central Sensitization Inventory.

**Table 1 medicina-59-01831-t001:** Characteristics of each variable in each cluster.

Variable	Overall (n = 40)	Cluster 1 (n = 19)	Cluster 2 (n = 11)	Cluster 3 (n = 10)
Age (years)	20.6 ± 0.6	20.5 ± 0.5	20.5 ± 0.5	21.0 ± 0.7
Height (cm)	165.4 ± 8.1	163.7 ± 8.1	167.4 ± 9.1	166.5 ± 7.6
Weight (kg)	55.7 ± 3.3	54.6 ± 10.9	56.3 ± 11.6	57.3 ± 7.1
BMI	20.3 ± 2.7	20.7 ± 2.0	20.0 ± 3.4	20.2 ± 2.7
VDT task time (hour/day)	3.4 ± 0.5	3.4 ± 0.5	3.4 ± 0.5	3.3 ± 0.7
Sleeping time (hour/day)	5.8 ± 1.2	5.8 ± 0.9	6.0 ± 0.9	5.9 ± 1.1
Physical exercise habit: mild load (times/week)	4.4 ± 2.3	4.1 ± 2.1	4.3 ± 2.7	5.0 ± 2.3
Physical exercise habit: moderate load (times/week)	1.8 ± 1.7	1.8 ± 2.0	1.6 ± 1.9	1.8 ± 1.3
Physical exercise habit: severe load (times/week)	0.6 ± 1.1	0.6 ± 1.2	0.6 ± 1.1	0.4 ± 0.7
Subjective stiffness	2.9 ± 2.9	5.3 ± 2.1	0.2 ± 0.6	1.4 ± 1.8
Pain	1.6 ± 2.1	3.3 ± 2.0	0 ± 0	0.2 ± 0.4
Muscle hardness (N)	34.7 ± 5.7	36.2 ± 3.9	38.9 ± 3.1	27.1 ± 3.1
PPT (N)	28.7 ± 11.3	31.1 ± 14.5	27.6 ± 7.5	25.3 ± 6.4
CVA (degree)	55.3 ± 6.1	55.5 ± 5.7	53.6 ± 6.3	56.9 ± 7.1
Spine angle (degree)	152.7 ± 6.6	152.6 ± 5.8	153.8 ± 6.8	151.6 ± 8.3
TDMS: pleasure	7.9 ± 4.5	7.6 ± 5.3	7.7 ± 4.4	8.5 ± 3.4
TDMS: arousal	−2.9 ± 3.4	−2.89 ± 3.4	−3.00 ± 4.5	−2.9 ± 2.6
TDMS: stability	5.6 ± 2.3	5.5 ± 2.5	5.5 ± 2.4	5.7 ± 2.1
TDMS: vitality	2.6 ± 3.2	2.6 ± 3.5	2.5 ± 3.8	2.8 ± 2.2
Distress	3.5 ± 2.3	4.1 ± 2.6	3.1 ± 1.9	2.7 ± 2.1
CSI	21.8 ± 11.5	27.9 ± 10.9	13.0 ± 9.4	19.7 ± 8.5
Fp1 alpha power (ln μV^2^)	1.4 ± 0.1	1.4 ± 0.1	1.5 ± 0.1	1.4 ± 0.1
Fp1 beta power (ln μV^2^)	1.6 ± 0.2	1.6 ± 0.2	1.7 ± 0.2	1.7 ± 0.3
Fp1 alpha/beta ratio	0.9 ± 0.1	0.9 ± 0.1	0.9 ± 0.1	0.9 ± 0.1
Fp2 alpha power (ln μV^2^)	1.4 ± 0.1	1.4 ± 0.1	1.5 ± 0.1	1.4 ± 0.1
Fp2 beta power (ln μV^2^)	1.6 ± 0.2	1.6 ± 0.1	1.7 ± 0.2	1.7 ± 0.3
Fp2 alpha/beta ratio	0.9 ± 0.1	0.9 ± 0.1	0.9 ± 0.1	0.9 ± 0.1
LF power (ln ms^2^)	2.6 ± 0.5	2.7 ± 0.4	2.5 ± 0.4	2.6 ± 0.5
HF power (ln ms^2^)	2.6 ± 0.5	2.8 ± 0.3	2.5 ± 0.3	2.6 ± 0.6
LF/HF ratio	−0.003 ± 0.3	−0.04 ± 0.4	0.03 ± 0.2	−0.01 ± 0.3

Data are expressed as mean ± SD. BMI, Body Mass Index; VDT, Visual Display Terminal; PPT, Pressure Pain Threshold; CVA, Craniovertebral Angle; TDMS, Two-dimensional Mood Scale; CSI, Central Sensitization Inventory; LF, Low Frequency; HF, High Frequency.

**Table 2 medicina-59-01831-t002:** Results of correlation analysis overall and by group by subjective stiffness.

Variable	Overall			Subjective Stiffness Group			Non-Subjective Stiffness Group
	Subjective Stiffness	Pain	Muscle Hardness	Subjective Stiffness	Pain	Muscle Hardness	Muscle Hardness
Age	−0.15	−0.03	−0.25	−0.19	0.04	−0.26	−0.26
Height	−0.05	−0.06	−0.01	0.14	0.22	−0.21	0.22
Weight	0.06	0.04	−0.12	0.32	0.35	−0.19	−0.04
BMI	0.10	−0.04	0.02	0.19	−0.17	0.25	−0.24
VDT task time	0.25	0.17	0.13	−0.18	−0.25	−0.13	0.46
Sleeping time	−0.01	−0.06	0.06	−0.07	−0.08	0.03	0.10
Physical exercise habit: mild load	−0.20	−0.14	−0.25	−0.24	−0.01	−0.30	−0.20
Physical exercise habit: moderate load	0.03	0.23	−0.20	0.22	0.65 **	−0.12	−0.33
Physical exercise habit: severe load	0.08	0.14	−0.02	0.11	0.33	−0.06	0.03
Subjective stiffness	1	0.71 **	0.18	1	0.41 *	0.45 *	1
Pain	0.71 **	1	0.16	0.41 *	1	0.35	0.31
Muscle hardness	0.18	0.16	1	0.45 *	0.35	1	−0.25
PPT	0.29	0.25	0.07	0.27	0.26	−0.05	−0.07
CVA	−0.05	−0.05	−0.14	0.02	−0.06	−0.05	−0.05
Spine angle	−0.12	−0.12	0.10	0.06	0.01	0.27	0.27
TDMS: pleasure	−0.16	−0.02	−0.14	−0.23	0.06	−0.06	−0.24
TDMS: arousal	0.08	0.07	−0.05	0.27	0.18	−0.12	0.02
TDMS: stability	−0.19	0.01	−0.09	−0.42 *	0.02	0.01	−0.21
TDMS: vitality	−0.05	0.08	−0.12	−0.02	0.20	−0.12	−0.13
Daily conscious psychological stress	0.44 **	0.27	0.22	0.50 *	0.17	0.13	0.35
CSI	0.59 **	0.48 **	0.06	0.42 *	0.15	0.07	0.07
Fp1 α power	−0.13	−0.23	0.05	−0.01	−0.12	−0.05	0.19
Fp1 β power	0.10	0.13	0.11	0.14	0.22	0.18	0.01
Fp1 α/β ratio	−0.22	−0.33 *	−0.09	−0.19	−0.38	−0.30	0.17
Fp2 α power	−0.13	−0.14	0.07	0.03	0.05	−0.02	0.20
Fp2 β power	0.08	0.09	0.11	0.16	0.2	0.17	0.01
Fp2 α/β ratio	−0.18	−0.18	−0.04	−0.18	−0.21	−0.23	0.16
LF power	0.15	0.03	0.04	0.24	0.1	0.18	−0.23
HF power	0.12	0.10	−0.06	0.38	0.34	0.24	−0.61
LF/HF ratio	0.03	−0.11	0.14	−0.25	−0.41	−0.10	0.43

** *p* < 0.01, * *p* < 0.05. BMI, Body Mass Index; VDT, Visual Display Terminal; PPT, Pressure Pain Threshold; CVA, Craniovertebral Angle; TDMS, Two-dimensional Mood Scale; CSI, Central Sensitization Inventory; LF, Low Frequency; HF, High Frequency.

## Data Availability

The data presented in this study are available upon request from the corresponding author. The data are not publicly available due to privacy and ethical restrictions.

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
