# Peer review of "Cluster Analysis of Subjective Shoulder Stiffness and Muscle Hardness: Associations with Central Sensitization-Related Symptoms"

_medicina, 2023, doi:10.3390/medicina59101831_

Round 1

Reviewer 1 Report

Authors conducted the study related “Cluster Analysis of Subjective Shoulder Stiffness and Muscle Hardness: Associations with Central Sensitization-Related Symptoms” to evaluate the associations among various factors including subjective shoulder stiffness, muscle hardness, and comprehensive factors which are yet to be fully understood. However, there are several shortcomings and areas for improvement in each section of the manuscript:

Comment 1: Abstract:

The abstract provides a good overview of the study but could be more concise. Some sentences are quite lengthy and could be broken down for better readability.

Comment 2: Introduction:

Authors provide extensive background information on shoulder stiffness, which is useful should be more concise. Authors are suggested to focus on why the study was conducted and address the specific research gaps. In addition, the hypotheses regarding the four subgroups of shoulder stiffness could be clarified and presented more explicitly.

Comment 3: Materials and Methods:

Authors are requested to provide STROBE check list and explain the points according to this list.

The study provides a thorough description of various objective measures (e.g., EEG, ANS, muscle hardness), it would be helpful to briefly explain why each of these measures was chosen and how they relate to the research questions.

Comment 4: Results:

 The results section is quite brief and should be expanded to include more detailed explanations and interpretations of the findings. Specifically, discuss the implications of the significant differences and correlations found.

Comment 5: Discussion:

The discussion section should provide a more in-depth interpretation of the results, including why certain correlations or differences were observed and their clinical or theoretical significance.

Comment 6: Authors mentioned several limitations in manuscript, but it would be more suitable to discuss these limitations and their potential impact on the study's findings and conclusions. Authors are requested to discuss the practical implications of the study's findings as well.

Comment 7:  It would be better if authors suggest the areas for future research based on the limitations and gaps identified in the current study.

Comment 8: Conclusion: Authors should summarize the key findings and their implications in a clear and concise manner.

Comment 9: The manuscript has several grammatical and language issues that are required to be improved. As careful proofreading and editing are essential to improve the clarity and readability of the text.

The manuscript has several grammatical and language issues that are required to be improved. As careful proofreading and editing are essential to improve the clarity and readability of the text.

Reviewer 2 Report

This observational study is proposed to verify the existence of subgroups of shoulder stiffness based on subjective symptoms and muscle hardness and to clarify the associated factors. It is well structured and the approaches are good.

For the evaluation of the VisionPose application (line 165), no reference is provided or whether this application has been used in other studies for the same purpose.

The values on the ordinate axis in figures 1 and 2 could be clearer.

Regarding the conclusions, do you see that there are any possible clinical implications? 
